behaviour/ecology/computational biology

division of labour, network analysis, community detection, social insect, age polyethism, monomorphic ant

# Bipartite network analysis of ant-task associations reveals task groups and absence of colonial daily activity

Haruna Fujioka[1,2], Yasukazu Okada[3] and Masato S. Abe[4]

[1]Graduate School of Arts and Sciences, the University of Tokyo, 3-8-1 Komaba, Meguro-ku, Tokyo, Japan
[2]Graduate School of Science, Osaka City University, 3-3-138 Sugimoto-cho, Sumiyoshi-ku, Osaka 558-8585, Japan
[3]Department of Biological Sciences, Tokyo Metropolitan University, 1-1 Minamiosawa, Hachioji, Tokyo, Japan
[4]Center for Advanced Intelligence Project, RIKEN, Nihonbashi 1-chome Mitsui Building, 1-4-1 Nihonbashi, Chuo-ku, Tokyo 103-0027, Japan

YO, 0000-0003-4528-2980; MSA, 0000-0002-0468-1179

Social insects are one of the best examples of complex self-organized systems exhibiting task allocation. How task allocation is achieved is the most fascinating question in behavioural ecology and complex systems science. However, it is difficult to comprehensively characterize task allocation patterns due to behavioural complexity, such as the individual variation, context dependency and chronological variation. Thus, it is imperative to quantify individual behaviours and integrate them into colony levels. Here, we applied bipartite network analyses to characterize individual-behaviour relationships. We recorded the behaviours of all individuals with verified age in ant colonies and analysed the individual-behaviour relationship at the individual, module and network levels. Bipartite network analysis successfully detected the module structures, illustrating that certain individuals performed a subset of behaviours (i.e. task groups). We confirmed age polyethism by comparing age between modules. Additionally, to test the daily rhythm of the executed tasks, the data were partitioned between daytime and nighttime, and a bipartite network was re-constructed. This analysis supported that there was no daily rhythm in the tasks performed. These findings suggested that bipartite network analyses could untangle complex task allocation patterns and provide insights into understanding the division of labour.

**Authors for correspondence:**
Haruna Fujioka
e-mail: fujioka.ha@gmail.com
Masato S. Abe
e-mail: masatoabee@gmail.com

# 1. Introduction

How the simple elements at lower levels can evolve to higher-level systems—e.g. the evolution of solitary individuals into social animal groups—through natural selection has been a central question in evolutionary biology [1,2]. Such systems often exhibit the division of labour through efficient task allocation, in which different elements perform different tasks [3]. It has been considered that the division of labour can be a primary advantage for social evolution [4–6]. Therefore, it is crucial to understand how the division of labour is achieved.

Social insects are one of the most sophisticated examples of division of labour as individuals exhibit reproductive division of labour between the queen(s) and workers, and various tasks are allocated among workers [5,7–9]. Individuals of many insect colonies are often specialized, so that workers engage in nursing, foraging, nest defence or cleaning. Task allocation is accomplished without any central control or hierarchal control [3,6,10,11]. Instead, local cues, such as interactions with others, play a key role [9,12–14]. Although an understanding of the proximate causes and ultimate consequences of task allocation is crucial, it is difficult to describe individual behaviours and quantify the task allocation patterns. This difficulty arises from the fact that there are a large number of individuals with different characteristics, such as morphology, age and spatial position in a colony [15–18]. Also, individual workers generally show flexibility in task performance [19–22]. Additionally, some typical behaviours can be decomposed into smaller components. For instance, brood care can be divided into nursing eggs, larvae and pupae. Therefore, methods to extract useful information from such complexity (i.e. many elements and high variation) and to integrate them to understand the complex social systems are strongly required.

Over the past two decades, network analysis has developed as a useful tool for analysing complex systems in various fields, including ecology, social science and animal behaviour [23–33]. The network perspective assumes that systems comprise nodes (i.e. elements) and links (i.e. the relationship between elements). The quantification of networks makes it possible to provide information on the structure of systems or characteristics of each individual within the network. Previous studies have revealed the structures of ant interaction networks [30,34,35], and there are some functional units within a colony that are based on tasks, such as foraging and brood care [17]. Additionally, it has been shown that these networks underlie the spreading dynamics of information, food or disease in a colony [25,31–34,36,37]. The described examples suggest that network analyses can provide information about the characteristics at various scales, from the individual to the colony level, and may provide new insights into understanding social insects.

Most network analyses in animal societies show relationships among the individuals within a group [17,35], although they can be applied to any elements and relationships. Previous studies have suggested that the task allocation patterns in social insects can be described as a bipartite network (figure 1), which has two node classes, and links are established only between nodes in different classes [38,39]. Workers are connected to the tasks they are engaged in, and tasks are linked to the workers who perform them [38,39]. Pasquaretta & Jeanson [39] quantified the degree of division of labour at the individual and colony levels by using an information theory approach. Additionally, they demonstrated that a community detection method for bipartite networks is an effective approach to determine the clusters of individuals that are engaged in similar subsets of tasks. However, bipartite network analyses have greater potential to quantify the task allocation patterns in terms of various levels (i) network level, (ii) cluster level and (iii) individual level. Moreover, the analysis also makes it possible (iv) to combine the properties in the network structure with biological features (e.g. diurnal activity, age). Here, we applied the following four new methods to characterize individual-behaviour relationships.

First, we applied nestedness to quantify the overlapping structure of the bipartite network as the index of network level [40,41]. In theoretical studies, the nested structure is generally related to the stability and persistence of the complex system [42–44]. In individual-behaviour networks, we might observe the overlapping structure of executed tasks because the overlapping structure reflects that generalists, which can emerge in small groups in particular [4], complement tasks in a colony [45,46]. When specialists that perform a certain task are lost unexpectedly in a colony with high nestedness, the generalists can supplement the task, leading to avoidance of critical function loss at the colony level. Therefore, nestedness could be a useful index for describing the robustness of colonies.

Second, we evaluated a module-module network (hereafter module network) to quantify the relationships between task groups. Although this idea of task networks was proposed approximately two decades ago [9,21,47], bipartite network analysis would offer a refined approach to construct task networks. As mentioned earlier, some behaviours can be decomposed into smaller components.

(*a*)

| | behaviour | | | | | | | |
|---|---|---|---|---|---|---|---|---|
| | brood-care day 1 | walking day 1 | inactive day 1 | foraging day 1 | brood-care day 2 | walking day 2 | inactive day 2 | foraging day 2 |
| Individual 1 | 2 | 0 | 6 | 0 | 2 | 0 | 6 | 0 |
| Individual 2 | 3 | 0 | 5 | 0 | 1 | 0 | 7 | 0 |
| Individual 3 | 6 | 2 | 0 | 0 | 6 | 2 | 0 | 0 |
| Individual 4 | 5 | 3 | 0 | 0 | 2 | 6 | 0 | 0 |
| Individual 5 | 0 | 7 | 0 | 1 | 0 | 6 | 0 | 2 |
| Individual 6 | 0 | 0 | 2 | 6 | 0 | 0 | 0 | 8 |
| Individual 7 | 0 | 0 | 0 | 8 | 0 | 0 | 0 | 8 |

(*b*)

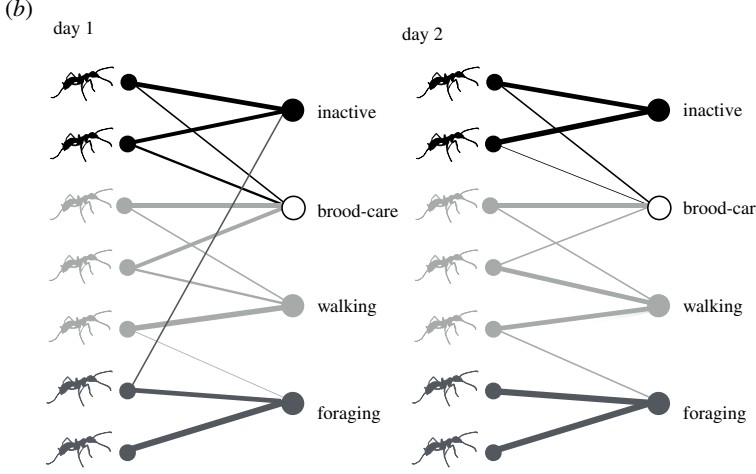

(*c*)

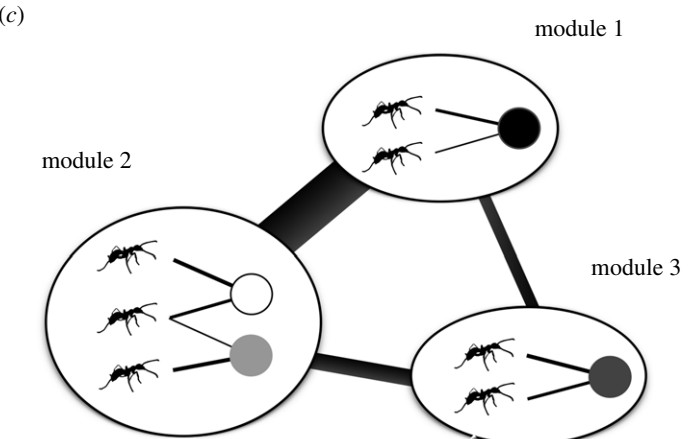

**Figure 1.** Individual-behaviour relationship as a bipartite network. An adjacency matrix for individual behaviour (*a*) can be described as a bipartite network with weights and information on day (*b*). The thickness of the links represents the weight of the interactions. Module structures can be detected by maximizing modularity (*c*). There were specialized individual-behaviour relationships in each module. The links between the modules represent the extent to which individuals within a module exhibited behaviour in the other modules.

Therefore, it is imperative to integrate such behavioural components into task groups. Once we identify task groups, we can consider the groups as new nodes at the module network level. This approach can quantify the relationship between task groups as a module network, that is, how task groups (modules) connect with each other (figure 1*c*). For example, workers' tasks could change from the

order of brood-care to forage [17,48]. Thus, workers rarely execute both brood care and forage. In this case, the module of brood care may be less connected with that of forage.

Third, we quantified the degree of specialization of each individual with consideration of module structures based on the idea that nodes with the same role should have similar topological properties [49,50]. Although previous studies on the division of labour have analysed the degree of specialization based only on the ratio of executed behaviours (e.g. Shannon entropy, $d'$, DOL) [39,51], the group structures were not incorporated into these indices. To consider the topological property for the degree of specialization, we applied the $c$-score, which was defined as among-module connectivity, (i.e. the level to which the individuals were linked to other modules) to evaluate the degree of specialization at the individual level. A low $c$-score indicates a high degree of specialization in the task group, and vice versa.

Finally, we proposed how to integrate other parameters with the network metrics. For example, individual characteristics, such as age or body size, are crucial for understanding the factors underlying task allocation [5]. Therefore, we investigated whether similarly aged individuals belonged to the same task groups. Additionally, chronological information can be incorporated in the analyses (e.g. year, season and time of day). Here, to investigate the daily rhythms of tasks, daytime (9:00–18:00) and night-time (21:00–6:00) data were separately analysed, and the network structures were compared between daytime and nighttime.

In the present study, we observed all colonial individuals with verified age in a ponerine ant (*Diacamma* sp.) and analysed individual-behaviour networks. We addressed three specific questions that are central to the study of colony organization in social insects: (i) What is the structure of individual-behaviour networks? (ii) How are the task groups distributed among workers? (iii) Does the age of the individual or the time of day affect the task group pattern? This study further applied network analysis to individual-behaviour networks, leading to a comprehensive approach to understanding task allocation in complex social systems.

# 2. Material and methods

## 2.1. Insects

The ponerine ant *Diacamma* sp. from Japan (the only species of the genus *Diacamma* found in Japan) is a monomorphic ant. The colony size of this ant species ranges from 30 to 200 workers. Young workers (typically 0–30 days of age) stay inside the nest and care for the brood (i.e. eggs, larvae and pupae), whereas workers older than 30 days start to forage outside the nest for various terrestrial invertebrates [52]. The lifespan of individuals is approximately one year in this focal ant [53].

## 2.2. Colony set-up

Five colonies of *Diacamma* sp. were excavated in Sueyoshi Park (Naha) and Kenmin-no mori (Onna), Japan, and [49] maintained in the laboratory in artificial plastic nests that were filled with moistened plaster as previously described [54]. The colonies were maintained at 25°C under a 16 L/8 D cycle (light phase, 07:00–23:00). By covering the top of the nest with a red film, the inside-nest environment was kept under dim red light. The reared colonies were fed chopped mealworms and crickets three times per week. The colonies consisted of a single mated female (gamergate) and workers. The colony sizes ranged from 38 to 167 individuals (including the queen); however, data could not be obtained for some individuals because of the loss of tags or death (electronic supplementary material, table S1). To understand the relationship between the role of individuals in a colony and age, we marked newly enclosed workers approximately once per month to record the age of each individual. Detailed information about the colonies is provided in electronic supplementary material, table S1.

## 2.3. Observation of behaviour

To maintain the unique identification of each individual, we attached 2D barcode tags (2.0 mm × 2.0 mm) to the individuals with instant glue (3 M Scotch, liquid glue multipurpose, Product No. 7004). Photos of the arena, including the colony and foraging area, were taken every second with a CCD camera (GS3-U3-123S6C-C, 3000 × 4000 pixels; FLIR Systems, Wilsonville, OR, USA) under constant dim red light. Experiment nests filled with moistened plaster (20.0 × 10.0 × 5.0 cm) had an entrance (diameter =

1.0 cm) to connect the foraging arena ($37.0 \times 25.0 \times 11.5$ cm). The details of the experimental set-up are provided in electronic supplementary material, figure S1. During the observation period, we fed crickets at random times and provided water at all times.

Following the identification of individual ants using a tag-based tracking system (BugTag, Robiotec Ltd.), we observed all individuals for 5 min every 3 h (a total of 8 daily observations) for 3 consecutive days by analysing the videos. The behaviour during each 5-min observation was classified as either walking (active), inactive (completely immobile for 5 min), egg care, larva care, pupa care and foraging (outside the nest) following a previous study [52]. In our definition, walking indicated an active state inside the nest but did not include engaging in brood care or foraging. It included self-grooming, allo-grooming, eating inside the nest, trophallaxis [55] and aggressive behaviour [30]. Inactive was defined as completely immobile during the 5-min observation. Therefore, we classified the behaviour as walking if an inactive ant started walking. Brood care included carrying the brood or performing brood grooming. Note that the colonies had different brood conditions, as shown in table 1. Foraging included walking, carrying food, eating food and drinking water outside the nest. If ants showed brood care during any part of the observation period, we classified the observation period as brood care. This classification was also applied to foraging. Note that there were no individuals showing both brood care and foraging during the 5 min observation.

## 2.4. Constructing bipartite networks

We constructed a bipartite network with weights for each colony from the behavioural data as defined above (figure 1a,b). The nodes represent ant individuals and behaviours, and the links represent the behaviours of individuals (figure 1b). Note that the behaviour data were separated by day (i.e. day 1, day 2 and day 3) to examine the consistency of patterns across different days (figure 1a). Connectance was defined as the ratio of the realized links to the possible links. The weights on the links in each day represented the number of executions of the behaviour (maximum 8 per day). The row and column of an adjacency matrix for a colony represented the individuals and behaviours (e.g. foraging on day 1, foraging on day 2, foraging on day 3, nursing on day 1), respectively (figure 1a). The detailed information about each network is shown in table 1.

## 2.5. Community detection in a bipartite network

To quantify the modules that were functional units from the individual-behaviour networks, we adopted community detection methods that can classify the node sets into groups by maximizing modularity [56,57]. Modularity indicates the degree of the link density within a given set of nodes and the link sparsity between nodes belonging to different sets. The maximized modularity provides the information on the extent to which the network has module-like structures, and the classification obtained from maximizing the modularity indicates the number of modules and the structure of the sub-networks. Note that the behaviours in a module obtained from this analysis can be interpreted as a 'task group'. In addition to the links, weights can provide important information about the extent to which an individual contributes to a certain task and reflect the roles of individuals within a colony. Hence, to assess the module structures, we used the modularity defined for bipartite networks with quantitative interaction strength (i.e. weights) [40]. The definition of the modularity QW for such networks was as follows:

$$Q_w = \frac{1}{M} \sum_{i=1}^{r} \sum_{j=1}^{s} (A_{ij} - E_{ij}) \delta(g_i, h_j) \,, \tag{2.1}$$

where $M$ is the total weight of the network, $A_{ij}$ is the $r \times s$ matrix including the number of behaviour executions and $E_{ij}$ is the expected value from a random network with the same degree distribution. $r$ and $s$ represent the number of rows (i.e. individuals) and columns (i.e. behaviours) of $A_{ij}$, respectively. For the weighted bipartite networks, $E_{ij}$ is $y_i z_j / M$, where $y_i$ and $z_j$ are the total weights of individual $i$ and behaviour $j$, respectively [40]. $\delta$ is a delta function that outputs 1 if $i$ and $j$ are in the same module ($g_i$ and $h_j$ represent module annotations to which $i$ and $j$ belong, respectively) and 0 if they are in different modules. Thus, when the links or weights within modules are larger than the expected values, $Q_W$ becomes larger.

The maximization of the modularity was conducted using the DIRTLPAwb + algorithm developed in [58]. This method was more robust than a previously proposed algorithm, QuanBio, for bipartite

**Table 1.** Colony information and the properties of network structure.

| colony ID | num. of ants | num. of behaviours × days | sum of weights | num. of links | num. of modules | connectance | modularity | normalized modularity | nestedness |
|---|---|---|---|---|---|---|---|---|---|
| a | 33 | 18 | 736 | 300 | 3 | 0.505 | 0.294* | 0.333 | 24.03* |
| b | 41 | 12 | 881 | 325 | 4 | 0.66 | 0.264* | 0.283 | 20.05 |
| c | 78 | 9 | 1609 | 524 | 3 | 0.746 | 0.245* | 0.333 | 21.71 |
| d | 104 | 15 | 2304 | 788 | 4 | 0.505 | 0.227* | 0.302 | 18.1 |
| e | 122 | 12 | 1999 | 791 | 3 | 0.54 | 0.293* | 0.262 | 20.62 |

networks with weights [58]. We ran the algorithm by using the computeModules function implemented in the 'bipartite' package v.2.11 of R 3.4.3. To compare the empirical results with those of random networks, we used null models that had the same marginal totals as the empirical models, but the number of links was variable (r2d null model). Moreover, to compare the modularities among networks, the value was normalized by the maximum value of modularity in the possible network configuration with the same total weight [39,58]. To do so, we used a specific function provided by a previous study [39] and finally obtained the normalized modularity, which ranged from 0 to 1. The *p*-value for modularity was calculated by identifying the position of empirical modularity in the distribution of the 1000 modularities obtained from the null model.

Once we detected the modules that contained certain individuals and behaviours, we then constructed the module networks. In most cases, there were some connections between modules, which suggested the relationships between functions. For example, individuals in a foraging module may be likely to enter the inactive states included in other modules and also unlikely to exhibit the behaviours included in the nursing module. To quantify these relationships, we calculated the mean total weights $q_{ij}$ of the links from the individuals belonging to module $i$ to tasks belonging to module $j$. Note that the value $q_{ii}$ represents a self-loop link from the individuals in module $i$ to the behaviour(s) in module $i$, showing the degree of specialization of task module $i$. Moreover, we analysed the types of individuals and behaviours that were included in each module as well as the relationship between the modules (figure 1*c*).

Then, to characterize the role of individuals in networks and modules, we calculated the *c*-score for each individual proposed by [41]. The *c*-score represents the degree of specialist-generalist in terms of the functional module. The definition of *c*-score $c_i$ for individual $i$ is as follows:

$$c_i = 1 - \sum_{j=1}^{N_M} \left(\frac{d_{ij}}{d_i}\right)^2,$$ (2.2)

where $d_i$ denotes the total weights of the links from individual $i$, $d_{ij}$ denotes the weights of links from individual $i$ to module $j$ and $N_M$ is the number of modules obtained from the modularity analysis. When the *c*-score is 0, the individual was engaged only in the behaviour of the module to which the individual belonged, suggesting that a smaller *c*-score represents a more specialized task choice. This value was similar to the participation coefficient in [49] but different from that in the presence of weights.

Moreover, we compared the *c*-score to the bipartite metrics $d'_{indiv}$ in a previous study [39] to confirm that the two metrics capture different aspects of the bipartite network structure. The *c*-score reflects how much an individual specializes in the task in the module to which it belongs. By contrast, the value of $d'_{indiv}$ indicates the degree of specialization by the original weighted adjacency matrix, which suggests that the value does not take into account the module structure [39,59]. The value $d'_{indiv}$ ranges from 0 (no specialization) to 1 (perfect specialist). The values of $d'_{indiv}$ were calculated by the function *dfun* in the package 'bipartite' in v.2.5 of R 3.4.3.

## 2.6. Nestedness

To calculate nestedness, we used the WNODF (Weighted Nestedness metric based on Overlap and Decreasing Fill) algorithm proposed in [60]. This method can quantify nestedness by incorporating the weights of links and produce correct results compared to WINE [60]. Three values, WNODF, WNODFc and WNODFr, were calculated using the nested function in the 'vegan' package v.2.5 of R 3.4.3. Then, we compared empirical values with those of null models [60] in which the marginal totals were equivalent to empirical values, but the number of links was variable. The *p*-value was calculated by identifying the position of the empirical data in the distribution of the 1000 values obtained from the null model. See the mathematical definition of nestedness in [43].

## 3. Results

We constructed bipartite networks with weights from the observation of each individual's behaviour for three days based on our behaviour criteria (see Methods). The number of individuals in each network, the number of behaviours, the number of links, and the sum of weights are shown in table 1. Figure 2 shows the visualized bipartite networks with weights with an alluvial diagram (https://rawgraphs.

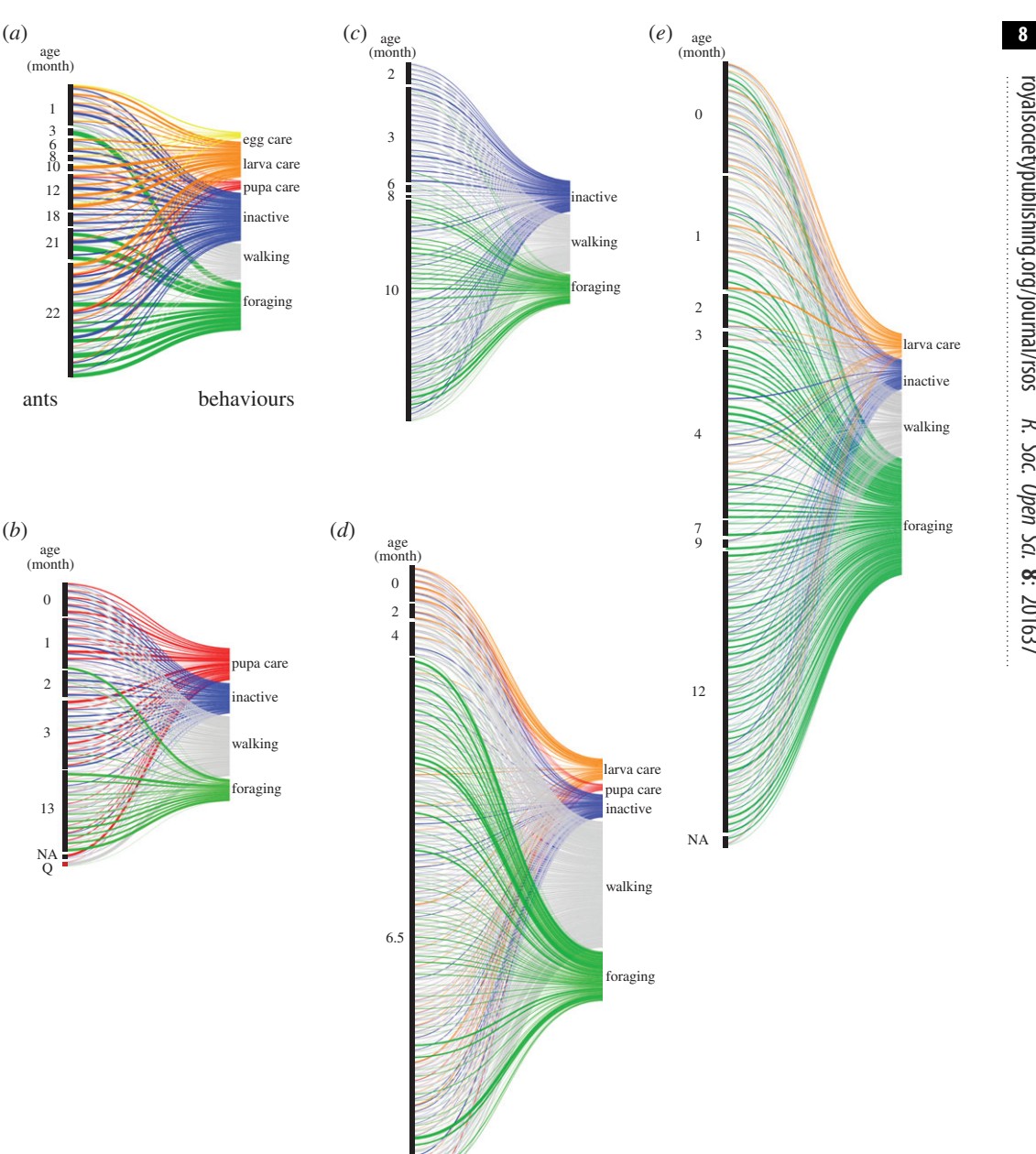

**Figure 2.** Task-individual networks of *Diacamma* sp. in five colonies (*a–e*). The left blocks are clusters of individual nodes depending on age. The right blocks are nodes of behaviour. The flows represent the links between individuals and behaviour. The numbers next to the left blocks represent age (month). The colours indicate behaviour. Each network was constructed from 3-day observation data, and the maximum weight of a link is 24. The width of each link is the number of instances of the behaviour. Nodes representing ant individuals are sorted by age (months), from youngest (top) to oldest (bottom). Q indicates queen (reproductive individual).

io/). The visualization showed that most individuals in the networks were not completely specialized to a certain behaviour for all colonies, that is, many individuals performed multiple behaviours.

## 3.1. Network-level metrics and module structures

First, we used four network-level properties: connectance, modularity, number of modules and nestedness [58,60]. One (colony a) of five colonies had a significantly nested structure (table 1). The comparison of modularity between the observed data and null models showed that significantly high module structures were detected in all networks (table 1). In addition, the normalized modularities ranged from 0.264 to 0.333. This suggested that individual-behaviour networks did not have complete module

structures. Namely, workers were not completely specialists because the normalized modularity for the case of the complete specialization should be one. Although task allocation in social insects is considered a specialization to a specific task, we found that individuals were not completely specialized throughout the day at the behavioural level, that is, several individuals showed multiple behaviours.

Second, we assessed the module structure of individual-behaviour networks based on the information obtained by maximizing modularity [39]. This analysis revealed the number of modules that could be considered task groups composed of individuals and behaviours. The results showed that the number of modules was similar to the number of behaviour types (shown in table 1). The modules were composed of one or two behaviour(s) except for in colony a (figure 3), suggesting that individual-behaviour networks had sub-group structures that were significantly related to the behavioural categories we defined. Each day, most behaviours were found in the same module, suggesting that the task allocation patterns were consistent over 3 days (figure 4).

## 3.2. Network of task groups (modules)

Next, the properties were quantified at the module level. After the community detection analysis, the relationship between task groups was visualized as a module network to show how task groups (modules) were connected with each other (figure 3). As mentioned above, individuals were not completely specialized to certain behaviours and executed multiple behaviours. Hence, there were weak connections between the modules. To quantify the relationships among the modules, we calculated the mean weights within and between each module and visualized it as a network of modules (figure 3). $q_{ii}$ means the mean weights on links from ant individuals belonging to module $i$ to behaviour belonging to module $i$, and $q_{ij}$ indicates the mean weights on the links from ant individuals belonging to module $i$ to behaviours belonging to module $j$. In other words, this analysis built a new network of task groups to inform how task groups interacted with each other.

The module size and the number of modules varied among colonies (figure 3). In three colonies, the walk module was strongly connected with other modules, and the links were one-sided towards the walk module (figure 3b–d), suggesting that individuals belonging to the non-walk module sometimes walked. This is not surprising because the walk module is a certain state to shift a behaviour (task). These results also suggested that the walking module could be a hub in the network (figure 3b–d).

In these colonies, which had modules of inactive ants, the arrows from inactive to non-walking modules (i.e. forage, brood cares) were thin, as indicated by the arrows from other modules to walking (figure 3b,c,e). Like the foraging and brood-care groups, the module of inactive ants can be treated as a distinct task group. The mean link weights from the foraging to brood-care modules tended to be small, suggesting that individuals were less likely to engage in both foraging and brood-care tasks (figure 3b,d,e).

## 3.3. Individual roles in the network

We then analysed the characteristics of the individuals. To evaluate the degree of specialization for each individual within the colony, the $c$-score (i.e. the extent to which the individuals were linked to other modules) was calculated (figure 4). A low $c$-score indicated strong task specialization, and vice versa. Three colonies (a, b and c) did not have significantly specialized task groups. In colonies d and e, there was a module with significantly low $c$-score (figure 4d,e). This result suggested that some workers specialized in walking and foraging, but these patterns between colonies were less consistent. Additionally, there was a significant correlation between the $c$-score and $d'_{indiv}$ ($r = -0.48$, $p < 0.01$, electronic supplementary material, figure S2). However, a perfect specialist in the task group (i.e. $c$-score = 0) was not identical to the perfect specialist identified from the information theoretic measure (i.e. $d' = 1$).

## 3.4. Temporal characteristics: daily rhythm

In general, ants show circadian activity rhythms, and our focal ant species are basically diurnal [61–63]. It is possible that the level of task performance would change between day and night. Therefore, we separated the data into daytime (9:00–18:00) and nighttime (21:00–6:00) and assessed the cluster structure of the task-individual networks. If the task allocation patterns had day and night cycles, there would be clusters of daytime and nighttime. The results showed that the detected clusters included both daytime and nighttime data, and the modules were separated by behaviour

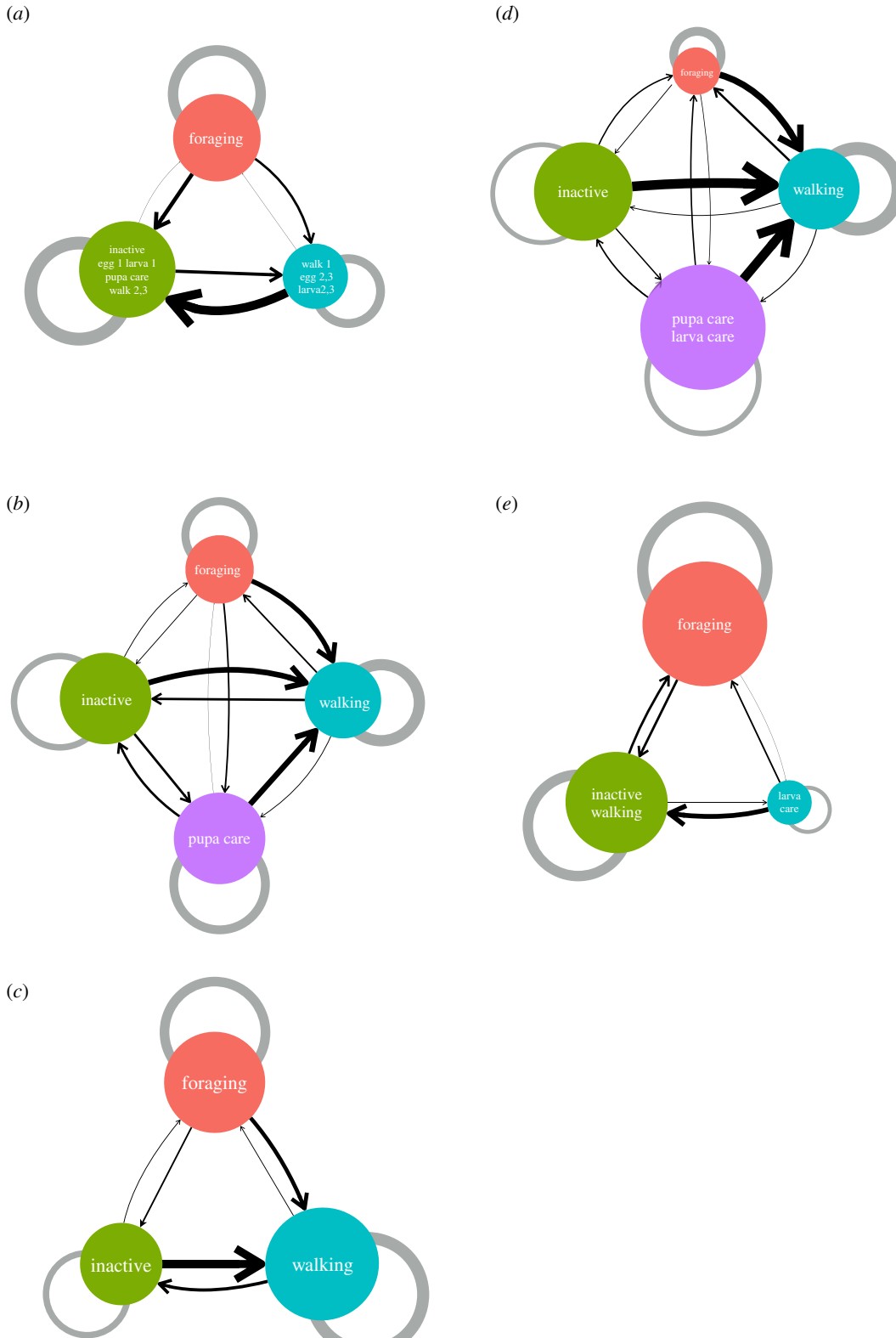

**Figure 3.** Networks of modules in 5 *Diacamma* sp. colonies (*a*–*e*). The node represents modules containing individuals and behaviour. The labels on the nodes indicate the behaviour contained in the module. The size of the nodes is weighted by the number of individuals belonging to the module. The links have a direction and width as the mean weight from ant individuals belonging to module $i$ to behaviour belonging to module $j$ ($q_{ij}$). The width of grey self-links indicates the mean weight from ant individuals belonging to module $i$ to behaviour belonging to the same module $i$ ($q_{ii}$). Day information was omitted when Days 1, 2 and 3 belonged to the same module.

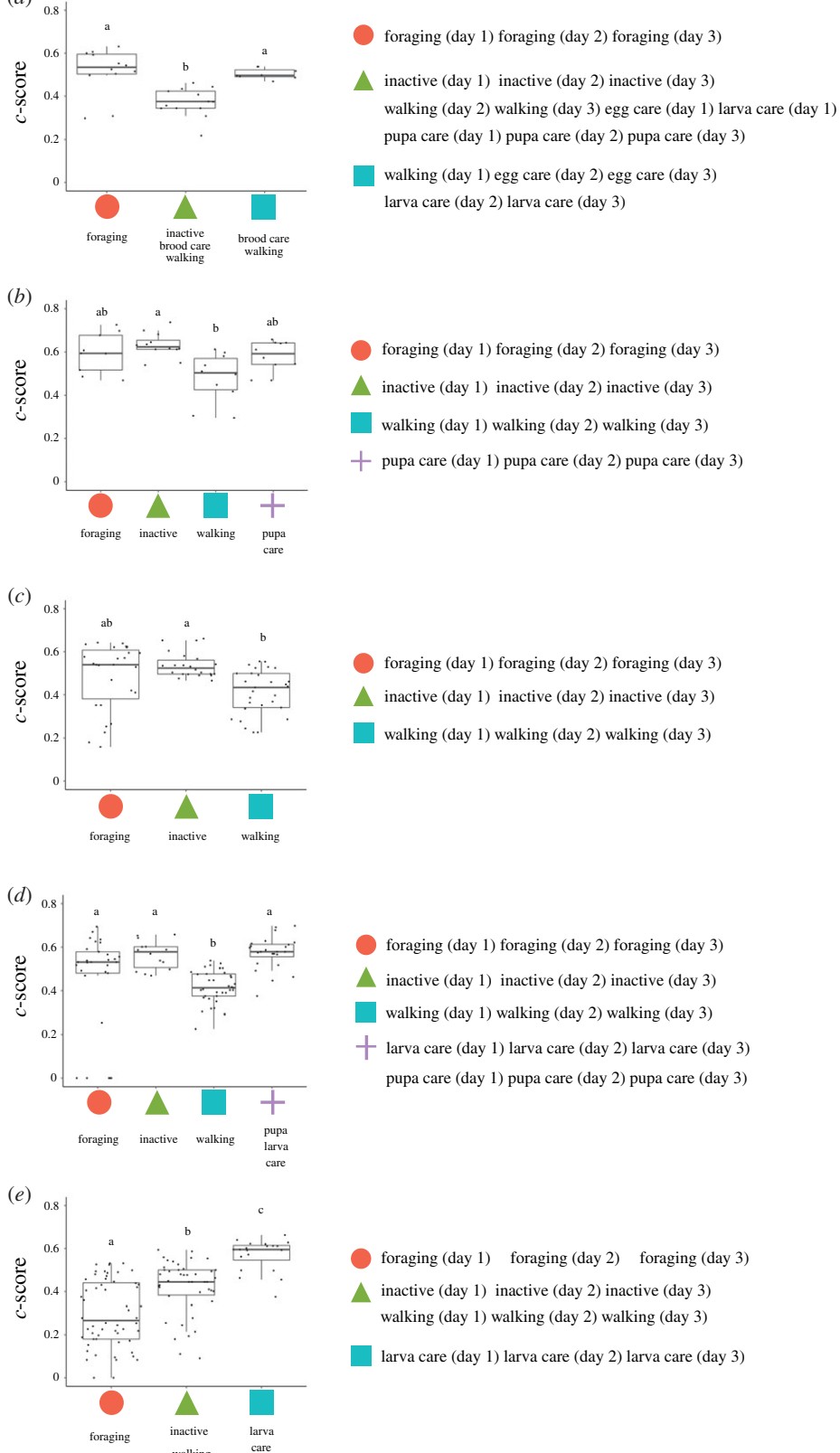

**Figure 4.** Relationship between modules and *c*-scores in 5 *Diacamma* sp. colonies (*a–e*). The *x*-axis represents each module, and the *y*-axis represents the *c*-score. A low *c*-score indicates strong specialization, and vice versa. The right panels indicate the behaviours (each day) that belong to each module. Different letters above the boxes indicate significant differences (by Steel–Dwass test, $p < 0.05$).

components (electronic supplementary material, figure S3), suggesting that the network structures were consistent throughout the day.

## 3.5. Relationship with age

The age-dependent changes in behaviours (i.e. forage, brood-care and inactive) were analysed by comparing the ages of individuals among modules (figure 5). The forage modules comprised old workers (figure 5a–d). On the other hand, the modules that included brood care were composed of young workers in colonies b, d and e (electronic supplementary material, figure S4b, d, e). From the viewpoint of the module structures, the task allocation pattern showed typical age polyethism in our focal species.

Moreover, we confirmed the pattern of age polyethism by plotting the proportion of executed behaviours as a function of age (electronic supplementary material, figure S4). Positive linear relationships were observed between forage and age in all colonies (a: slope = 0.0004, $R^2 = 0.12$; b: slope = 0.0008, $R^2 = 0.40$; c: slope = 0.001, $R^2 = 0.24$; d: slope = 0.001, $R^2 = 0.095$; e: slope = 0.0005, $R^2 = 0.09$; electronic supplementary material, figure S4a), while there was a negative relationship between brood care and age except for colony a (b: slope = −0.0004, $R^2 = 0.12$; d: slope = −0.0003, $R^2 = 0.12$; e: slope = −0.0003, $R^2 = 0.15$; electronic supplementary material, figure S4b). Although there were large variations between individuals under the same age class, the task allocation patterns showed typical age polyethism, which was consistent with a previous study on *Diacamma* sp. [64]. We also showed the proportion of inactive as the number of inactivity per the 24 observations. In three colonies, the proportion of inactive was significantly negatively related to age (a: slope = −0.0002, $R^2 = 0.12$; b: slope = −0.003, $R^2 = 0.03$; c: slope = −0.0005, $R^2 = 0.08$; electronic supplementary material, figure S4c), suggesting that the young workers tended to be inactive.

# 4. Discussion

To understand the complex task allocation patterns in insect societies, network analysis is helpful for quantifying the characteristics of task allocation. In this study, we recorded the behaviours of all individuals in five ant colonies of *Diacamma* sp. and investigated the individual-behaviour networks using the bipartite network approach. We found a non-nested structure and determined the characteristics of module networks, including the inactive groups. Moreover, we detected consistent module structures in classifying task groups throughout a day (i.e. no daily cycle) and three consecutive days (i.e. no daily changes).

Network analysis can summarize the task allocation patterns as various values of a network-level index. The result of a significantly large modularity suggested the presence of the module structures, which were comprised of some individuals and a few behaviours. However, the normalized modularity largely deviated from 1, suggesting that there were no complete specializations and that individuals were loosely linked to each behaviour. We found no significant nested structure in most colonies (table 1), suggesting that a specific individual did not cover the tasks that other individuals executed. Interestingly, the smallest colony had a significant nested structure (table 1). This structure may reflect the notion that generalists can emerge to accomplish all tasks when colony size is small [4]. It is possible that young (small) colonies are vulnerable to disturbance. If a specialized individual is dead, the task cannot be executed, which can lead to the critical function loss. Thus, a high nestedness structure might be a reasonable strategy for maintaining a colony's productivity. Conversely, the benefits of specialization may offset the decline in robustness in large colonies. Further studies are needed in order to investigate the relationship between the network structure (i.e. nestedness) and the colony size [18].

The number of tasks within a colony is one of the fundamental factors in task allocation research. While many studies have focused on only a few prominent tasks, such as foraging, building and brood care (e.g. [11,65,66]), high specialization has been observed in other tasks, such as guarding [48,67], grooming [68,69] and inactive [70]. At this point, how many tasks exist in insect societies is still not completely understood. This seemingly simple question is not easy to answer because the demand for tasks can change depending on the context. A community detection method can be valid for the classification of task groups (i.e. modules) [39]. In this study, we found that the number of modules differed among colonies (figure 3). We also found not only the module of forage and brood care but also the module of walk and inactive. It suggested that the inactive state could be categorized

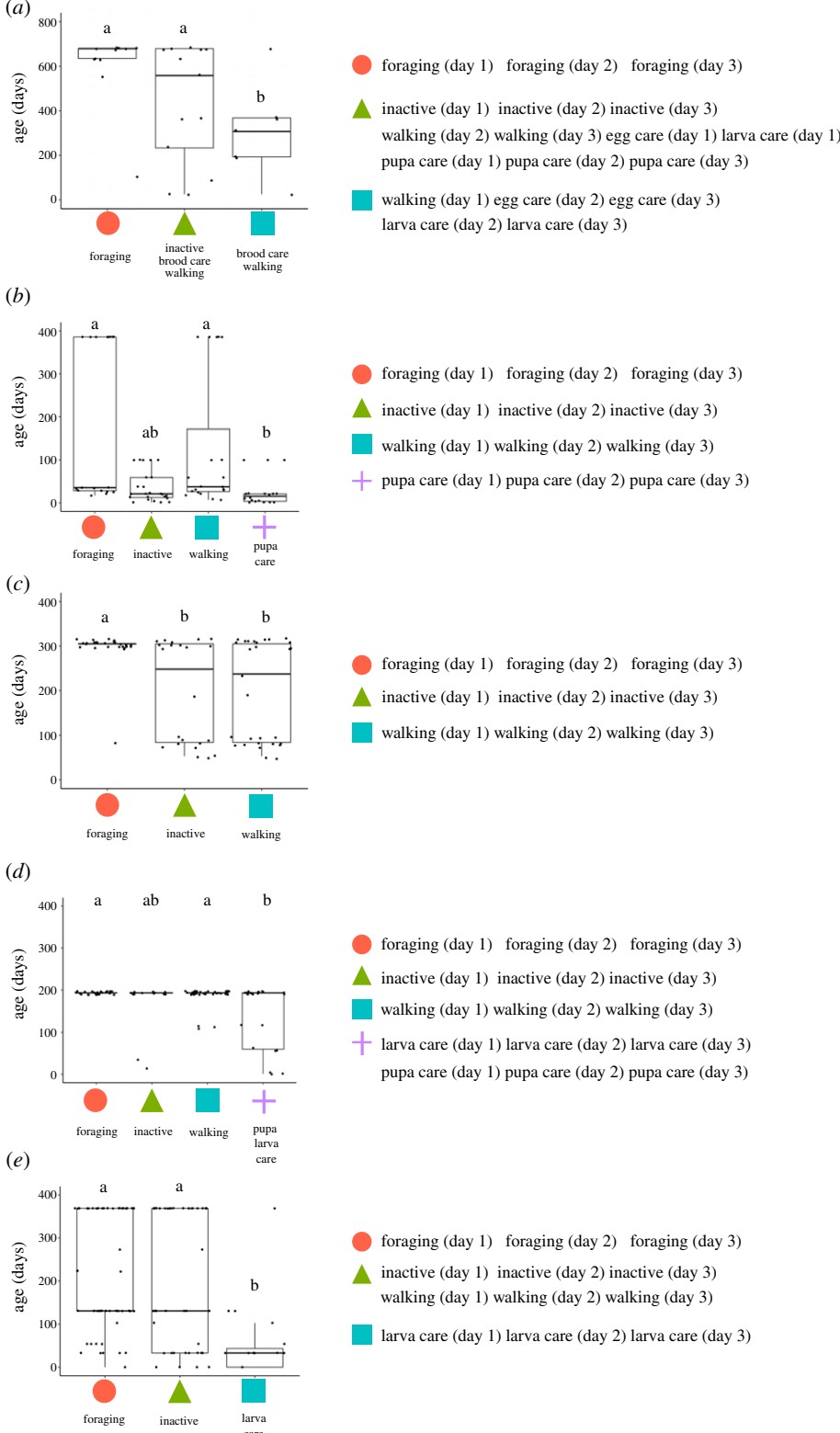

**Figure 5.** Ant age in each module of *Diacamma* sp. colony (*a*–*e*). The *x*-axis represents each module, and the *y*-axis represents age (days). Black dots are individual values. The right panels indicate the behaviours (each day) that belong to the modules. Different letters (e.g. a, b, c) above the box indicate significant differences by Steel–Dwass test (*p* < 0.05).

as one of the task groups, which was consistent with the previous study [70]. In our definition, walking included several behaviours, such as grooming, trophallaxis and dominance behaviours. Therefore, it is possible that there were several task groups in the 'walk' module. A more detailed observation should be incorporated in future studies.

Other clustering methods (e.g. hierarchical clustering analysis) can also classify workers into separate groups [70]. However, there are at least two advantages of maximizing modularity instead of other methods. First, we could determine the number of modules based on the modularity criteria. Second, comparing empirical networks with null models, which are constructed from a randomly shuffled adjacency matrix, made it possible to infer the behavioural rules underlying the empirical task allocation patterns. In our analysis, we used the null model in which the behaviour of each individual was executed as a proportion of the total number of executed behaviours (see method 2.5). In other words, each individual chooses a behaviour depending only on the work demands. The results showed that there were significant differences between the empirical and null networks (table 1, modularity). Therefore, we confirmed that individual ants determined their behaviour depending not only on the task demands but also on other factors, including internal states, experiences or interactions with other individuals.

We quantified the $c$-score, which was the degree of specialization based on module structures obtained through community detection. Using this approach, we evaluated individual roles with respect to their position within and among modules. Previous studies have quantified the degree of specialization from the original weighted adjacency matrix (e.g. $d'$ [59] and $DOL$ [51]). These metrics do not take the relationships that connect across modules into account. As modules are considered as functional task groups if modules are not taken into account, important properties could be overlooked. Some complete specialists with a $c$-score $= 0$ have $d'_{indiv} < 1$ (electronic supplementary material, figure S2). When individuals in a module engage in only tasks in the module, the $c$-score becomes zero. Hence, the $c$-score reflects the degree of specialization in a functional module. For example, the $c$-score of an individual in a nursing module represents how much the individual specializes in a function, including nursing eggs, larva and pupa. By contrast, the value $d'$ represents the degree of specialization for each behaviour. These metrics provided different information about the level of specialization. Here, we would like to emphasize that the combination of several metrics is needed to understand the whole aspect of task allocation at the individual level, as pointed out by a previous study [39].

Once we obtained individual-level metrics, it was possible to answer further questions, such as what kind of individuals belonged to the module by comparing the scores between modules (figure 4) and what the relationship was with other individual traits, such as age. Additionally, this method enabled us to perform inter-species as well as inter-colony comparisons. As shown in figure 2, since task allocation patterns were complex, it would be difficult to compare them without such metrics.

Behaviour can change with time, such as daily rhythms. We analysed the daily changes by dividing the data into daytime and nighttime (electronic supplementary material, figure S3). We found that there was no daytime and nighttime module structure, suggesting that the task allocation pattern was consistent throughout the day. Interestingly, solitary ants show circadian activity rhythms even under constant light conditions [61–63]. Previous studies have shown that nurse ants take care of the brood throughout the day [61,62]. Moreover, *Diacamma* sp. is known to occasionally forage at night according to field observations [71]. Foragers might be active during the night as long as the temperature is suitable for foragers. Our results supported these previous findings by providing evidence of a non-circadian pattern in task execution. In addition to the daily cycle, temporal information, such as their experiences and previous state, might affect decision making in individuals. This assumption might be addressed in future studies, exploring the temporal network analyses that have ordered temporal data.

For social insect colonies, an increasing number of studies have indicated that worker inactivity is common (reviewed in [72]). In our focal ant, some individuals specialized in inactivity (figure 3). Although the existence of inactive individuals has been documented across many social insect species, we know very little about the variations between individuals. The simplest hypothesis for inactive (lazy) ants is immaturity or senescence (proposed by [73,74]). Young workers may be less active due to their still-developing physiology; on the other hand, older workers may be less active due to degraded physiology. Here, we tested this hypothesis. Our data showed that young workers tended to be inactive; however, there were differences between colonies. Although age could be related to inactivity, our results suggested that immaturity was related to inactivity; however, ant inactivity was also colony dependent.

Assessing how the quantified patterns emerged based on decision making in individuals is our next question for understanding task allocation. Previous studies have shown that the interactions between individuals and division of labour can have substantial interplay [14,75]. Thus, it is imperative to integrate individual-behaviour networks revealed here into the interaction networks between individuals. Our results on the module networks implied that not only the interactions between individuals within a module but also the interactions between individuals in different modules with significant links were important. Further studies are needed to examine how interactions affect individual decision making and subsequent task allocation patterns. One possible way is to analyse

mathematical models to construct the patterns of the networks from rules at individual levels [3,75,76]. The network perspective can capture complex phenomena composed of many elements and extract important information at multiple levels. The individual-task relationships in mammal and even human societies [77] are crucial for understanding the division of labour in complex societies. The methodology we introduced here may be applied to a wide range of individual-behaviour networks.

Data accessibility. All data are available in the electronic supplementary material.

Authors' contributions. H.F., Y.O. and M.S.A. designed the study. H.F. conducted the experiment. H.F. and M.S.A. conducted the analysis. H.F. and M.S.A. wrote the first draft of the manuscript. All authors contributed to the final version of the manuscript, gave final approval for publication, and agree to be held accountable for the work performed therein.

Competing interests. We declare we have no competing interests.

Funding. This study was funded by JSPS KAKENHI, with grant nos JP18J13369, JP20J01766 to H.F., JP17K19381, JP18H04815 to Y.O. and JP15H06830 to M.S.A., and the Grant in Scientific Research on Innovation Areas 'Integrative Research toward Elucidation of Generative Brain Systems for Individuality' JP17H05938 and JP19H04913 from MEXT to Y.O. This work was also supported by the Sasakawa Scientific Research Grant from the Japan Science Society to H.F.

Acknowledgements. Many thanks to Hakataya S. for helping with the ant-keeping work. We also thank Uematsu J. for kindly supporting us with ant excavation and Hamano M. for helping with ant observations.

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
