## [Reviewer comments · Royal Society Open Science]

Review History

RSOS-200576.R0 (Original submission)

Review form: Reviewer 1

Is the manuscript scientifically sound in its present form?

No

Are the interpretations and conclusions justified by the results?

Yes

Is the language acceptable?

Yes

Do you have any ethical concerns with this paper?

No

Have you any concerns about statistical analyses in this paper?

No

Recommendation?

Major revision is needed (please make suggestions in comments)

Comments to the Author(s)

In the paper, the authors aim to use a network approach to quantify the intensity of division of labor in ant colonies. I have several concerns with the manuscript.

The approach claimed by the authors has already been used in a previous study (Pasquaretta and Jeanson 2018). It is perfectly normal to build on previous work, but it must be properly credited. The paper by Pasquaretta and Jeanson is only briefly mentioned in the present manuscript and I regret that the authors did not give more credit to this previous study. For instance, the authors state "we propose that a community detection algorithm can be valid for the classification of task groups" (line 84-85) or "this study shows that network analysis ... can lead to a comprehensive approach for understanding task allocation..." (lines 107-108). The way in which the sentences are formulated is ambiguous to say the least, again this was developed in the paper by Pasquaretta and Jeanson 2018. If the authors feel that they are contributing something new, it would have been useful to show how their metrics capture essential aspects of division of labor that were not captured by previously developed measures.

My recommendation would be that the authors reframe their paper. They should focus on the biological issue and propose a more comprehensive study of the division of labor in their ant species. They should use measures already developed, possibly supplemented by additional measures (e.g. nestedness) provided that their addition is well explained and justified.

The introduction is a bit confusing and the objectives of the article are unclear. The articulation between the methodological approach and the experimental results could be greatly improved. As things stand now, it is difficult to grasp the big picture of the manuscript.

The authors indicate that they performed 5-min visual observations every three hours (lines 143). It is not clear how the authors scored behavior if the ant changed activity (e.g., became inactive and then walked) during these 5 minutes. Or did the authors use scan sampling? This needs to be explained.

The authors argued that the "resolution for constructing networks in division of labour is 24 hours" (line 160-161). I do not agree with this argument. In my opinion, there is no such rule and the appropriate "resolution" depends on the species of ant, the age of the colony and many other factors.

Line 204: I don't understand what means "if i equals j ". If i equals j , there is only one module, right? It is unclear.

Lines 220-236: this paragraph is confusing and needs to be detailed more comprehensively. The authors argue that nestedness may be the most useful index (line 224), but unfortunately this section is poorly written. As it stands, it is unclear what this index brings to the understanding of the division of labor.

Figure 3: In the legend, the authors indicate that "the width of self-links indicate the weight from ant belonging to module i to behavior belonging to the same module". This is confusing. I don't understand how there can be self-loop for modules that comprised only one behavior (e.g. forage, inactivity). Again, this needs to be explained.

Line 149, 350: typos. Grooming not glooming

Review form: Reviewer 2

Is the manuscript scientifically sound in its present form?

Yes

Are the interpretations and conclusions justified by the results?

Yes

Is the language acceptable?

No

Do you have any ethical concerns with this paper?

No

Have you any concerns about statistical analyses in this paper?

No

Recommendation?

Major revision is needed (please make suggestions in comments)

Comments to the Author(s)

The authors present a method for analyzing the division of labor in an ant colony as a bipartite network between workers and the tasks (really, task groups) that they perform. Given the increasing interest in social networks in animal societies, particularly in social insect colonies and in relation to division of labor, this paper has the potential to offer a nice contribution to the growing work on animal societies and its intersection with network science. However, I feel the paper in the current form suffers from a lack of clarity and focus making it difficult for the reader to both understand and take away a clear message from the paper.

First, on the clarity side, the writing needs significant reworking. Throughout the paper there are noticeable grammar mistakes. While these should be corrected for the sake of writing itself, these mistakes also make it much harder for the reader to follow the paper. For example:

- The first sentence of the introduction has so many commas and dependent clauses that it makes it hard for the reader. An em dash here and throughout the paper could help make dependent clauses within a sentence less disruptive: "How the simple elements at a lower level can evolve to higher-level systems – e.g., solitary individuals into social animal groups – through natural selection has been a central interest of evolutionary biology."

- The sentence on line 63 starting with "Previous studies..." does not have nouns that match. Two are plural nouns and one is singular: "Previous studies...reveal the individual variation in the number of links,...the module,...and the spreading dynamics". Individual variation in the module? Individual variation in the spreading dynamics (which are group-level phenomena)? Additionally, the dependent clause in the middle of the sentence that starts with "which" again makes the sentence hard to decipher.

- The sentence on line 168 has a verb that does not match the nouns "To quantify the modules which is a functional units" -> "To quantify the modules that are functional units".

These are just some examples, but a careful reworking of the grammar and syntax throughout will greatly improve the clarity. Additionally, there are some parts about the methods that are not entirely clear, but seem very important for the paper. For example:

- Line 203-204: It is not clear how "if i equals j , the value represents the degree of specialisation of the task module". Please consider explaining further.

Second, on the focus side, I feel it is unclear if the paper is presenting a new method or is examining known phenomena (e.g., age-based division of labor) in a new species. Initially the paper presents a new method and spends much detail explaining it and exploring what insights it gives into ant colony organization. However, figure 5 seems entirely different and instead examines age-based division of labor, absent of the new method. As a reader I am left wondering then if this paper is about the new method or about insights into this specific species? If it is the former, the paper should focus on establishing the method and how the insights it provides are different from other established methods. For example:

- How do the division of labor metric you use – c-score – compare to other commonly used division of labor metrics, notably the entropy-based score introduced by Gorelick, et al. 2004? Are the values you are seeing indicating something different about the nature of the division of labor in the colony that would be missed by using the entropy metric?

- How is your method different from the bipartite approach presented in Pasquaretta and Jeanson (2018)? I understand you are not setting the j nodes to be distinct tasks and are instead allowing multiple tasks to occupy a single node j , but that should be made much clearer. At first approach, the method presented in this paper seemed redundant to Pasquaretta and Jeanson (2018), so more time engaging that method and discussing the merits of your approach would go a long way in helping the reader understand your approach and its use. Ultimately, the readers will be very eager to understand how this method can help them gain more knowledge about their study system and when exactly one should use it.

If the paper's focus is about gaining insight into the specific species in the paper, then using a novel method when it might not be necessary seems a bit confusing and makes it harder to determine how these patterns really compare to other species. I think the authors should carefully consider which type of paper they prefer and restructure it accordingly.

Lastly, much emphasis in the introduction and discussion of the paper is placed on interactions between individuals, yet your method does not deal with social interactions between individuals. While you do construct an individual-task network, that seems fundamentally different than a social network (i.e., individual-individual network). If you wish to keep the emphasis on inter-individual interactions, then more work should be done discussing how this method provides insights into those dynamics of a colony. Do the results from your method reveal something unconsidered in normal social network analysis? Do your results confirm a prediction from theoretical models of interactions and task allocation? I suggest further engagement with empirical (e.g., Mersch, Crespi, & Keller 2013; Naug 2009) and theoretical work (e.g., Naug & Gadagkar 1999; Tokita & Tarnita 2020) examining the role of social interactions in division of labor and social insect networks. You do not need to cite the specific papers I listed, but I hope they at least serve as a helpful starting point in the search for further literature in this area.

I apologize for such a long message, but I wanted to provide detailed feedback to the authors to ensure I could be helpful. The good news to the authors is that the scientific core of this paper is sound and of interest, but it will require a significant update to the text of the paper to ensure readers can engage with it to the level it deserves.

Decision letter (RSOS-200576.R0)

Dear Dr Abe,

Manuscript ID RSOS-200576 entitled "Bipartite network analysis of task-ant associations reveals task groups and absence of colonial diurnal activity" which you submitted to Royal Society Open Science, has been reviewed. The comments from reviewers are included at the bottom of this letter.

In view of the criticisms of the reviewers, the manuscript has been rejected in its current form. However, a new manuscript may be submitted which takes into consideration these comments.

Please note that resubmitting your manuscript does not guarantee eventual acceptance, and that your resubmission will be subject to peer review before a decision is made.

Your resubmitted manuscript should be submitted by 24-Nov-2020. If you are unable to submit by this date please contact the Editorial Office.

on behalf of Dr Oliver Schülke (Associate Editor) and Kevin Padian (Subject Editor)
openscience@royalsociety.org

Associate Editor Comments to Author (Dr Oliver Schülke):

Dear Dr. Abe,

As the associate editor handling your submission I have now received comments from two expert referees. Based on these comments I will recommend to my editor to reject the manuscript. I agree with the reviewers though that the scientific core of the paper can make a valuable contribution to the literature and, therefore, suggest to allow re-submission of a thoroughly revised version. The important task will be to revisit the literature starting from but not limited to the papers identified the reviewers. Depending on whether the main contribution concerns a methodological advancement or the biology of the species studied, you will have to more clearly

develop exactly how the new method advances our analytical toolkit beyond what we already have or how our understanding of the species' biology or the phenomenon of division of labor shall be changed by your study. Reader comprehension of the writing will be enhanced by using shorter sentences with simple structure and careful revision of the grammar.

I hope that you will decide to resubmit to RSOS and look forward to reading the new manuscript.

With my best regards,
Oliver Schülke

Reviewers' Comments to Author:

Reviewer: 1

Comments to the Author(s)

In the paper, the authors aim to use a network approach to quantify the intensity of division of labor in ant colonies. I have several concerns with the manuscript.

The approach claimed by the authors has already been used in a previous study (Pasquaretta and Jeanson 2018). It is perfectly normal to build on previous work, but it must be properly credited. The paper by Pasquaretta and Jeanson is only briefly mentioned in the present manuscript and I regret that the authors did not give more credit to this previous study. For instance, the authors state "we propose that a community detection algorithm can be valid for the classification of task groups" (line 84-85) or "this study shows that network analysis ... can lead to a comprehensive approach for understanding task allocation..." (lines 107-108). The way in which the sentences are formulated is ambiguous to say the least, again this was developed in the paper by Pasquaretta and Jeanson 2018. If the authors feel that they are contributing something new, it would have been useful to show how their metrics capture essential aspects of division of labor that were not captured by previously developed measures.

My recommendation would be that the authors reframe their paper. They should focus on the biological issue and propose a more comprehensive study of the division of labor in their ant species. They should use measures already developed, possibly supplemented by additional measures (e.g. nestedness) provided that their addition is well explained and justified.

The introduction is a bit confusing and the objectives of the article are unclear. The articulation between the methodological approach and the experimental results could be greatly improved. As things stand now, it is difficult to grasp the big picture of the manuscript.

The authors indicate that they performed 5-min visual observations every three hours (lines 143). It is not clear how the authors scored behavior if the ant changed activity (e.g., became inactive and then walked) during these 5 minutes. Or did the authors use scan sampling? This needs to be explained.

The authors argued that the "resolution for constructing networks in division of labour is 24 hours" (line 160-161). I do not agree with this argument. In my opinion, there is no such rule and the appropriate "resolution" depends on the species of ant, the age of the colony and many other factors.

Line 204: I don't understand what means "if i equals j ". If i equals j , there is only one module, right? It is unclear.

Lines 220-236: this paragraph is confusing and needs to be detailed more comprehensively.

The authors argue that nestedness may be the most useful index (line 224), but unfortunately this section is poorly written. As it stands, it is unclear what this index brings to the understanding of the division of labor.

Figure 3: In the legend, the authors indicate that "the width of self-links indicate the weight from ant belonging to module i to behavior belonging to the same module". This is confusing. I don't understand how there can be self-loop for modules that comprised only one behavior (e.g. forage, inactivity). Again, this needs to be explained.

Line 149, 350: typos. Grooming not glooming

Reviewer: 2

Comments to the Author(s)

The authors present a method for analyzing the division of labor in an ant colony as a bipartite network between workers and the tasks (really, task groups) that they perform. Given the increasing interest in social networks in animal societies, particularly in social insect colonies and in relation to division of labor, this paper has the potential to offer a nice contribution to the growing work on animal societies and its intersection with network science. However, I feel the paper in the current form suffers from a lack of clarity and focus making it difficult for the reader to both understand and take away a clear message from the paper.

First, on the clarity side, the writing needs significant reworking. Throughout the paper there are noticeable grammar mistakes. While these should be corrected for the sake of writing itself, these mistakes also make it much harder for the reader to follow the paper. For example:

- The first sentence of the introduction has so many commas and dependent clauses that it makes it hard for the reader. An em dash here and throughout the paper could help make dependent clauses within a sentence less disruptive: "How the simple elements at a lower level can evolve to higher-level systems – e.g., solitary individuals into social animal groups – through natural selection has been a central interest of evolutionary biology."
- The sentence on line 63 starting with "Previous studies..." does not have nouns that match. Two are plural nouns and one is singular: "Previous studies...reveal the individual variation in the number of links,...the module,...and the spreading dynamics". Individual variation in the module? Individual variation in the spreading dynamics (which are group-level phenomena)? Additionally, the dependent clause in the middle of the sentence that starts with "which" again makes the sentence hard to decipher.
- The sentence on line 168 has a verb that does not match the nouns "To quantify the modules which is a functional units" -> "To quantify the modules that are functional units".

These are just some examples, but a careful reworking of the grammar and syntax throughout will greatly improve the clarity. Additionally, there are some parts about the methods that are not entirely clear, but seem very important for the paper. For example:

- Line 203-204: It is not clear how "if i equals j , the value represents the degree of specialisation of the task module". Please consider explaining further.

Second, on the focus side, I feel it is unclear if the paper is presenting a new method or is examining known phenomena (e.g., age-based division of labor) in a new species. Initially the paper presents a new method and spends much detail explaining it and exploring what insights

it gives into ant colony organization. However, figure 5 seems entirely different and instead examines age-based division of labor, absent of the new method. As a reader I am left wondering then if this paper is about the new method or about insights into this specific species? If it is the former, the paper should focus on establishing the method and how the insights it provides are different from other established methods. For example:

- How do the division of labor metric you use—c-score—compare to other commonly used division of labor metrics, notably the entropy-based score introduced by Gorelick, et al. 2004? Are the values you are seeing indicating something different about the nature of the division of labor in the colony that would be missed by using the entropy metric?
- How is your method different from the bipartite approach presented in Pasquaretta and Jeanson (2018)? I understand you are not setting the j nodes to be distinct tasks and are instead allowing multiple tasks to occupy a single node j , but that should be made much clearer. At first approach, the method presented in this paper seemed redundant to Pasquaretta and Jeanson (2018), so more time engaging that method and discussing the merits of your approach would go a long way in helping the reader understand your approach and its use. Ultimately, the readers will be very eager to understand how this method can help them gain more knowledge about their study system and when exactly one should use it.

If the paper's focus is about gaining insight into the specific species in the paper, then using a novel method when it might not be necessary seems a bit confusing and makes it harder to determine how these patterns really compare to other species. I think the authors should carefully consider which type of paper they prefer and restructure it accordingly.

Lastly, much emphasis in the introduction and discussion of the paper is placed on interactions between individuals, yet your method does not deal with social interactions between individuals. While you do construct an individual-task network, that seems fundamentally different than a social network (i.e., individual-individual network). If you wish to keep the emphasis on inter-individual interactions, then more work should be done discussing how this method provides insights into those dynamics of a colony. Do the results from your method reveal something unconsidered in normal social network analysis? Do your results confirm a prediction from theoretical models of interactions and task allocation? I suggest further engagement with empirical (e.g., Mersch, Crespi, & Keller 2013; Naug 2009) and theoretical work (e.g., Naug & Gadagkar 1999; Tokita & Tarnita 2020) examining the role of social interactions in division of labor and social insect networks. You do not need to cite the specific papers I listed, but I hope they at least serve as a helpful starting point in the search for further literature in this area.

I apologize for such a long message, but I wanted to provide detailed feedback to the authors to ensure I could be helpful. The good news to the authors is that the scientific core of this paper is sound and of interest, but it will require a significant update to the text of the paper to ensure readers can engage with it to the level it deserves.

Author's Response to Decision Letter for (RSOS-200576.R0)

See Appendix A.

RSOS-201637.R0

Review form: Reviewer 2

Is the manuscript scientifically sound in its present form?

Yes

Are the interpretations and conclusions justified by the results?

Yes

Is the language acceptable?

Yes

Do you have any ethical concerns with this paper?

No

Have you any concerns about statistical analyses in this paper?

No

Recommendation?

Accept with minor revision (please list in comments)

Comments to the Author(s)

Overall I think this paper is much improved. It is clear that the authors are building on Pasquaretta and Jeanson by first examining nestedness in the bipartite individual-task network and then expanding the bipartite method by (a) considering task-task module networks and including it in measurement of individual-level specialization, and (b) considering individual differences (in age, body size, etc.) within this bipartite network.

Some of the findings in this paper are very interesting. The ability of your metric to detect possible signs of system robustness (absence/presence of nestedness) and to detect functional specialization (lines 385-386) are very nice from a complex systems perspective.

Thus, I would suggest engaging a bit more with complex systems literature (either the portion focused on social insects or the broader literature), since you clearly touch on these themes but mostly in passing. Your metrics seems to offer interesting insights into these aspects of colony structure, and you could well lean into it a bit more.

You may also wish to engage with (or at least acknowledge) some of the older social insect literature that looked at task networks but constructed them largely manually (e.g., Deborah Gordon's earlier work on task flow networks in harvester ants, or the figure in Fewell 2003 in *Science*). I see some similarities in your network module approach, although yours is obviously far more modern.

Unfortunately, there are still some clarity issues with the writing. As you'll see below there are still portions of the paper where it is hard for the reader to clearly follow the logic and message of the paper. My stylistic suggestions do not need to be followed but instead are intended to point out where I found it hard to follow and provide a possible way to improve (not because I believe my way is best but simply to try to be helpful). More time dedicated to truly polishing the writing will bring this paper into final form!

INTRODUCTION

The introduction now makes it clear that this paper will make methodological contributions to the field. That said, the new sections that were added were still a bit hard to parse. For example, line 74-79 has an abrupt transition from discussing the gap in the Pasquaretta and Jeanson method that still needs to be filled to "Also, we focused on a network structure and incorporating other biological features (e.g., diurnal activity, age)." The use of "also" here is confusing relative to the previous sentence; you haven't discussed what you have done up until this point so "also we focused" is confusing. The end of this paragraph is key for the reader to understand what the gap is and how you intend to fill it before discussing what exactly you did in the following paragraphs. Please revise this carefully to help the reader along.

The following four new paragraphs where you describe specifically the four core things you are building upon the bipartite method can be a bit hard to follow. As of now it felt a bit like you are listing a bunch of things you will do/discuss in this paper but it's not clear why that's the case. Maybe if you could provide a roadmap (e.g., a numbered list of the four main things you're doing) to the reader before these paragraphs where you succinctly describe the four ways in which you will be building upon the Pasquaretta and Jeanson method, it will be much easier to follow. It doesn't have to be a roadmap, but consider now that unless the reader hops from topic sentence to topic sentence in this section, one can get a bit lost in the discussion and lose sight of the exact novelty you are bringing to the table.

METHODS

Lines 229-232: I think you could explain more about why exactly you want to compare your c-score metric with d'. You mention that you are going to compare your metric to d', but why? As of now it just seems an arbitrary comparison, so please make clear to the reader why you wish to compare it so that the later results showing the difference between the two are much clearer. Are you comparing it because you believe the two metrics will capture different aspects of division of labor/bipartite network structure? Are you comparing it because your work is expanding on the study that introduced d' so you want to use it as a baseline?

DISCUSSION

Lines 374-378: This paragraph seems to be a result and should probably be entirely/mostly moved to the results section. It mostly seems to be describing an empirical finding and not actually adding new discussion/insight to the finding.

Lines 385-386: "The fact that some complete specialists with c-score = 0 have $d'_{\text{indiv}} < 1$ (figure S2) reflects that the c-score can identify specialists from a functional point of view." In my opinion this is a key finding of your method and should be highlighted more. You are showing how your method and metrics gain a new insight into division of labor that is missed by other measurements.

As with the above lines, in general in your discussion, I think you could lead more with a key finding or point and discuss that instead of summarizing what you did in your study (which you have already done in the methods and results section!). You have some interesting findings that deserve highlighting and discussion but some of them are buried within your discussion paragraphs instead of leading them! For example, it seems the key topic in paragraph 2 of the discussion is your findings on nestedness, group size, and its implications for robustness, but it is sort of buried later in the paragraph. You could advocate more clearly in your paragraph that your approach gives insights into the organization of colonies that may be missed without your approach. I haven't heard many others discuss how the nestedness of task structure may change with group size, adding to your understanding of scaling effects in colonies. Very interesting!

Decision letter (RSOS-201637.R0)

Dear Dr Abe

On behalf of the Editors, we are pleased to inform you that your Manuscript RSOS-201637 "Bipartite network analysis of ant-task associations reveals task groups and absence of colonial daily activity" has been accepted for publication in Royal Society Open Science subject to minor revision in accordance with the referees' reports. Please find the referees' comments along with any feedback from the Editors below my signature.

Please submit your revised manuscript and required files (see below) no later than 7 days from today's (ie 16-Nov-2020) date. Note: the ScholarOne system will 'lock' if submission of the revision is attempted 7 or more days after the deadline. If you do not think you will be able to meet this deadline please contact the editorial office immediately.

on behalf of Dr Oliver Schülke (Associate Editor) and Kevin Padian (Subject Editor)
openscience@royalsociety.org

Associate Editor Comments to Author (Dr Oliver Schülke):

Dear Dr. Abe,
as the associate editor handling your submission, I have now received comments from one of your previous reviewers. Based on these comments and my own reading of the manuscript, I will suggest to my editor-in-chief accept your paper for publication pending minor revisions. Your reviewer has made some very helpful and concrete comments on how to further improve the quality of your contribution. I agree with the reviewer that the minor changes will help to more closely guide the reader and highlight the specific novelty of your work.

We are looking forward to receiving a revised version at your earliest convenience.
 With kind regards,
 Oliver Schülke

Reviewer comments to Author:
 Reviewer: 2

Comments to the Author(s)

Overall I think this paper is much improved. It is clear that the authors are building on Pasquaretta and Jeanson by first examining nestedness in the bipartite individual-task network and then expanding the bipartite method by (a) considering task-task module networks and including it in measurement of individual-level specialization, and (b) considering individual differences (in age, body size, etc.) within this bipartite network.

Some of the findings in this paper are very interesting. The ability of your metric to detect possible signs of system robustness (absence/presence of nestedness) and to detect functional specialization (lines 385-386) are very nice from a complex systems perspective.

Thus, I would suggest engaging a bit more with complex systems literature (either the portion focused on social insects or the broader literature), since you clearly touch on these themes but mostly in passing. Your metrics seems to offer interesting insights into these aspects of colony structure, and you could well lean into it a bit more.

You may also wish to engage with (or at least acknowledge) some of the older social insect literature that looked at task networks but constructed them largely manually (e.g., Deborah Gordon's earlier work on task flow networks in harvester ants, or the figure in Fewell 2003 in *Science*). I see some similarities in your network module approach, although yours is obviously far more modern.

Unfortunately, there are still some clarity issues with the writing. As you'll see below there are still portions of the paper where it is hard for the reader to clearly follow the logic and message of the paper. My stylistic suggestions do not need to be followed but instead are intended to point out where I found it hard to follow and provide a possible way to improve (not because I believe my way is best but simply to try to be helpful). More time dedicated to truly polishing the writing will bring this paper into final form!

INTRODUCTION

The introduction now makes it clear that this paper will make methodological contributions to the field. That said, the new sections that were added were still a bit hard to parse. For example, line 74-79 has an abrupt transition from discussing the gap in the Pasquaretta and Jeanson method that still needs to be filled to "Also, we focused on a network structure and incorporating other biological features (e.g., diurnal activity, age)." The use of "also" here is confusing relative to the previous sentence; you haven't discussed what you have done up until this point so "also we focused" is confusing. The end of this paragraph is key for the reader to understand what the gap is and how you intend to fill it before discussing what exactly you did in the following paragraphs. Please revise this carefully to help the reader along.

The following four new paragraphs where you describe specifically the four core things you are building upon the bipartite method can be a bit hard to follow. As of now it felt a bit like you are listing a bunch of things you will do/discuss in this paper but it's not clear why that's the case. Maybe if you could provide a roadmap (e.g., a numbered list of the four main things you're doing) to the reader before these paragraphs where you succinctly describe the four ways in which you will be building upon the Pasquaretta and Jeanson method, it will be much easier to

follow. It doesn't have to be a roadmap, but consider now that unless the reader hops from topic sentence to topic sentence in this section, one can get a bit lost in the discussion and lose sight of the exact novelty you are bringing to the table.

METHODS

Lines 229-232: I think you could explain more about why exactly you want to compare your c-score metric with d'. You mention that you are going to compare your metric to d', but why? As of now it just seems an arbitrary comparison, so please make clear to the reader why you wish to compare it so that the later results showing the difference between the two are much clearer. Are you comparing it because you believe the two metrics will capture different aspects of division of labor/bipartite network structure? Are you comparing it because your work is expanding on the study that introduced d' so you want to use it as a baseline?

DISCUSSION

Lines 374-378: This paragraph seems to be a result and should probably be entirely/mostly moved to the results section. It mostly seems to be describing an empirical finding and not actually adding new discussion/insight to the finding.

Lines 385-386: "The fact that some complete specialists with c-score = 0 have $d'_{\text{indiv}} < 1$ (figure S2) reflects that the c-score can identify specialists from a functional point of view." In my opinion this is a key finding of your method and should be highlighted more. You are showing how your method and metrics gain a new insight into division of labor that is missed by other measurements.

As with the above lines, in general in your discussion, I think you could lead more with a key finding or point and discuss that instead of summarizing what you did in your study (which you have already done in the methods and results section!). You have some interesting findings that deserve highlighting and discussion but some of them are buried within your discussion paragraphs instead of leading them! For example, it seems the key topic in paragraph 2 of the discussion is your findings on nestedness, group size, and its implications for robustness, but it is sort of buried later in the paragraph. You could advocate more clearly in your paragraph that your approach gives insights into the organization of colonies that may be missed without your approach. I haven't heard many others discuss how the nestedness of task structure may change with group size, adding to your understanding of scaling effects in colonies. Very interesting!

===PREPARING YOUR MANUSCRIPT===

===PREPARING YOUR REVISION IN SCHOLARONE===

<https://royalsociety.org/journals/authors/author-guidelines/#data>. You should ensure that

you cite the dataset in your reference list. If you have deposited data etc in the Dryad repository, please only include the 'For publication' link at this stage. You should remove the 'For review' link.

Author's Response to Decision Letter for (RSOS-201637.R0)

See Appendix B.

Decision letter (RSOS-201637.R1)

Dear Dr Abe,

It is a pleasure to accept your manuscript entitled "Bipartite network analysis of ant-task associations reveals task groups and absence of colonial daily activity" in its current form for publication in Royal Society Open Science.

on behalf of Dr Oliver Schülke (Associate Editor) and Kevin Padian (Subject Editor)
openscience@royalsociety.org

Appendix A

Dear Dr. Oliver Schülke (Associate Editor),

We would like to resubmit the revised version of our manuscript entitled “Bipartite network analysis of ant-task associations reveals task groups and absence of colonial daily activity”.

We deeply thank the editor and two reviewers for thoroughly reviewing our manuscript and making many insightful comments.

According to the comments from the reviewers, we have revised the manuscript.

The major changes are as follows:

- 1) The main contribution of our study is a methodological advancement. Therefore, we have reviewed the previous bipartite approach in the division of labour (lines 71-119) and clarified the following four points of new method advances:
 - 1) nestedness (lines 80-88); 2) the visualization of a module network (lines 89-96); 3) the *c*-score, which is the degree of specialization for each individual in consideration of module structures (lines 97-105); and 4) the connection of age with the individual’s role within a network (lines 106-112).
- 2) We have polished the English language with the support of a native English speaker.

Our point-by-point responses to each of the reviewers’ comments are detailed below. The modified parts of the manuscript are marked in blue.

Sincerely yours,

Masato S. Abe

RIKEN Center for Advanced Intelligence Projects,
Nihonbashi 1-chome Mitsui Building, 15th floor,
1-4-1 Nihonbashi, Chuo-ku, Tokyo 103-0027, Japan
Tel: +81-048-467-3627
masato.abe@riken.jp

Response to Reviewer #1

Comment 1: In the paper, the authors aim to use a network approach to quantify the intensity of division of labor in ant colonies. I have several concerns with the manuscript.

The approach claimed by the authors has already been used in a previous study (Pasquaretta and Jeanson 2018). It is perfectly normal to build on previous work, but it must be properly credited. The paper by Pasquaretta and Jeanson is only briefly mentioned in the present manuscript and I regret that the authors did not give more credit to this previous study. For instance, the authors state "we propose that a community detection algorithm can be valid for the classification of task groups" (line 84-85) or "this study shows that network analysis ... can lead to a comprehensive approach for understanding task allocation..." (lines 107-108). The way in which the sentences are formulated is ambiguous to say the least, again this was developed in the paper by Pasquaretta and Jeanson 2018. If the authors feel that they are contributing something new, it would have been useful to show how their metrics capture essential aspects of division of labor that were not captured by previously developed measures. My recommendation would be that the authors reframe their paper. They should focus on the biological issue and propose a more comprehensive study of the division of labor in their ant species. They should use measures already developed, possibly supplemented by additional measures (e.g. nestedness) provided that their addition is well explained and justified.

The introduction is a bit confusing and the objectives of the article are unclear. The articulation between the methodological approach and the experimental results could be greatly improved. As things stand now, it is difficult to grasp the big picture of the manuscript.

Response: Thank you for your helpful comments. As you noted, we used the community detection method that has been previously applied by Pasquaretta and Jeanson (2018). However, we also used some new methods to capture individual-task relationships. To clarify our contributions, we have added explanations of both the previous studies (lines 71-74) and listed the four methodological approaches that have been newly applied: 1) nestedness (lines 80-88); 2) the module network (lines 89-96); 3) the *c*-score, which is the new index of specialization for each individual in consideration of module structures (lines 97-105); and 4) the connection of age with the individual's role within a network (lines 106-112). We apologize that our explanations were not enough and correct in the previous manuscript.

Comment 2: The authors indicate that they performed 5-min visual observations every three hours (lines 143). It is not clear how the authors scored behavior if the ant changed activity (e.g., became inactive and then walked) during these 5 minutes. Or did the authors use scan sampling? This needs to be explained.

Response: Thank you for the suggestion. If an inactive ant became active (walked), we defined the behaviour as walking. In our definition, inactivity was defined as being completely immobile during a 5-min observation (lines 158-159, 162-164). We observed the behaviours of all the individuals by watching the video repeatedly. We have added explanations about these methods to the manuscript (lines 152-154).

Comment 3: The authors argued that the "resolution for constructing networks in division of labour is 24 hours" (line 160-161). I do not agree with this argument. In my opinion, there is no such rule and the appropriate "resolution" depends on the species of ant, the age of the colony and many other factors.

Response: We apologize for our incorrect explanation. We used 24 hours of resolution because we are interested in the behavioural consistency across days. We thought that a period of 24 hours was one of the possible resolutions (time windows) used for constructing networks. However, we agree with your suggestion. Therefore, we have omitted this sentence.

Comment 4: Line 204: I don't understand what means "if i equals j ". If i equals j , there is only one module, right? It is unclear.

Response: Thank you for pointing this out. We meant to refer to the mean weights from the individuals in module i to the behaviours in the same module i . We have improved this sentence by using the variable q_{ii} (lines 216-217).

Comment 5: Lines 220-236: this paragraph is confusing and needs to be detailed more comprehensively.

The authors argue that nestedness may be the most useful index (line 224), but unfortunately this section is poorly written. As it stands, it is unclear what this index brings to the understanding of the division of labor.

Response: Thank you for the suggestion. We have added explanations of why nestedness is useful for understanding the division of labour in the Introduction section of the manuscript (lines 80-88).

Comment 6: Figure 3: In the legend, the authors indicate that "the width of self-links indicate the weight from ant belonging to module i to behavior belonging to the same module". This is confusing. I don't understand how there can be self-loop for modules that comprised only one behavior (e.g. forage, inactivity). Again, this needs to be explained.

Response: The modules contain both individual(s) and behaviour(s). Therefore, the width of the self-loop represents the mean weights (q_{ii}) from the individuals belonging to module i to the behaviours

belonging to the same module i . We have improved the description within the manuscript (lines 216-217).

Comment 7: Line 149, 350: typos. Grooming not glooming

Response: We have corrected the typos (lines 157, 352).

Response to Reviewer #2

Comment 1: The authors present a method for analyzing the division of labor in an ant colony as a bipartite network between workers and the tasks (really, task groups) that they perform. Given the increasing interest in social networks in animal societies, particularly in social insect colonies and in relation to division of labor, this paper has the potential to offer a nice contribution to the growing work on animal societies and its intersection with network science. However, I feel the paper in the current form suffers from a lack of clarity and focus making it difficult for the reader to both understand and take away a clear message from the paper.

Response: We strongly appreciate for your careful reading and many insightful comments on our manuscript.

Comment 2: First, on the clarity side, the writing needs significant reworking. Throughout the paper there are noticeable grammar mistakes. While these should be corrected for the sake of writing itself, these mistakes also make it much harder for the reader to follow the paper. For example:

- The first sentence of the introduction has so many commas and dependent clauses that it makes it hard for the reader. An em dash here and throughout the paper could help make dependent clauses within a sentence less disruptive: "How the simple elements at a lower level can evolve to higher-level systems—e.g., solitary individuals into social animal groups—through natural selection has been a central interest of evolutionary biology."

- The sentence on line 63 starting with "Previous studies..." does not have nouns that match. Two are plural nouns and one is singular: "Previous studies...reveal the individual variation in the number of links,...the module,...and the spreading dynamics". Individual variation in the module? Individual variation in the spreading dynamics (which are group-level phenomena)? Additionally, the dependent clause in the middle of the sentence that starts with "which" again makes the sentence hard to decipher

- The sentence on line 168 has a verb that does not match the nouns "To quantify the modules which is a functional units" -> "To quantify the modules that are functional units".

These are just some examples, but a careful reworking of the grammar and syntax throughout will greatly improve the clarity

Response: We strongly appreciate your careful reading of and many insightful comments about our manuscript.

We have corrected the points you suggested as follows:

- We have used an em dash (lines 38-39).
- We have divided this thought into two sentences (lines 58-63).
- We have changed the text to the following: "To quantify the modules that are functional units" (line 178).
- We have had the manuscript re-edited by a professional language editing company to address all language problems.

Comment 3: Additionally, there are some parts about the methods that are not entirely clear, but seem very important for the paper. For example:

- Line 203-204: It is not clear how "if i equals j , the value represents the degree of specialisation of the task module". Please consider explaining further.

Response: Thank you for pointing this out. We meant to refer to the mean weights from the individuals in module i to the behaviours in the same module i . We have improved this sentence by using the variable q_{ii} (lines 216-217).

Comment 4: Second, on the focus side, I feel it is unclear if the paper is presenting a new method or is examining known phenomena (e.g., age-based division of labor) in a new species. Initially the paper presents a new method and spends much detail explaining it and exploring what insights it gives into ant colony organization. However, figure 5 seems entirely different and instead examines age-based division of labor, absent of the new method. As a reader I am left wondering then if this paper is about the new method or about insights into this specific species? If it is the former, the paper should focus on establishing the method and how the insights it provides are different from other established methods.

If the paper's focus is about gaining insight into the specific species in the paper, then using a novel method when it might not be necessary seems a bit confusing and makes it harder to determine how these patterns really compare to other species. I think the authors should carefully consider which type of paper they prefer and restructure it accordingly.

Response: Thank you for your suggestion. As you have pointed out, our focus was not clear in the previous manuscript. This study newly applied four methodological approaches to the ant-behaviour network. We have revised the Introduction section to clarify these methodological novelties (lines 71-112).

We agree that the results of Figure 5 extend beyond our focus. In this study, we proposed how to integrate characteristics other than the ant-behaviour network, such as age and time, with

network metrics (lines 106-112). Therefore, we have omitted Figure 5 (age-polytheism) from the main text and moved it to the supplementary materials section.

Comment 5: For example:

- How do the division of labor metric you use—*c*-score—compare to other commonly used division of labor metrics, notably the entropy-based score introduced by Gorelick, et al. 2004? Are the values you are seeing indicating something different about the nature of the division of labor in the colony that would be missed by using the entropy metric?

Response: The *c*-score represents the degree of specialization for each individual in consideration of the module structures. The module structures were not incorporated into the previous metrics (e.g., the Shannon entropy and the *DOL*). We followed the idea that nodes with the same role should have similar topological properties (Guimera 2005). The *c*-score reflects the degree of specialization in each task group rather than in each behaviour. We have added an explanation of the *c*-score in the Introduction section (lines 97-105). Moreover, we have added the result of a comparison between the previous metric *d'* and the *c*-score (Figure S2). We have also discussed the fact that these metrics provide different information on the level of specialization and that a combination of several metrics is needed to understand complex task allocation patterns (lines 379-389, figure S2).

Comment 6: How is your method different from the bipartite approach presented in Pasquaretta and Jeanson (2018)? I understand you are not setting the *j* nodes to be distinct tasks and are instead allowing multiple tasks to occupy a single node *j*, but that should be made much clearer. At first approach, the method presented in this paper seemed redundant to Pasquaretta and Jeanson (2018), so more time engaging that method and discussing the merits of your approach would go a long way in helping the reader understand your approach and its use. Ultimately, the readers will be very eager to understand how this method can help them gain more knowledge about their study system and when exactly one should use it.

Response: We have clarified the differences between our approach and that of Pasquaretta and Jeanson (2018) in the Introduction section (lines 71-76, 99-101). We used a method (i.e., modularity) that has been previously applied in an earlier study. We have also corrected this point within the text.

Comment 7: Lastly, much emphasis in the introduction and discussion of the paper is placed on interactions between individuals, yet your method does not deal with social interactions between individuals. While you do construct an individual-task network, that seems fundamentally different than a social network (i.e., individual-individual network). If you wish to keep the emphasis on inter-individual interactions, then more work should be done discussing how this method provides insights into those dynamics of a colony. Do the results from your method reveal something unconsidered in

normal social network analysis? Do your results confirm a prediction from theoretical models of interactions and task allocation? I suggest further engagement with empirical (e.g., Mersch, Crespi, & Keller 2013; Naug 2009) and theoretical work (e.g., Naug & Gadagkar 1999; Tokita & Tarnita 2020) examining the role of social interactions in division of labor and social insect networks. You do not need to cite the specific papers I listed, but I hope they at least serve as a helpful starting point in the search for further literature in this area.

Response: We agree with the suggestion that our previous manuscript could possibly confuse readers. Although we thought that the interactions between individuals were possible mechanisms underlying the task-ant network structure, we focused on an individual-behaviour network in this study. In our future research, we would like to address the interesting issue of how to connect individual-behaviour networks with individual-individual networks (lines 416-424). These points have been clarified in the updated manuscript (lines 416-424).

Comment 8: I apologize for such a long message, but I wanted to provide detailed feedback to the authors to ensure I could be helpful. The good news to the authors is that the scientific core of this paper is sound and of interest, but it will require a significant update to the text of the paper to ensure readers can engage with it to the level it deserves.

Response: We are deeply grateful to you for taking the time to provide valuable comments. Your comments have encouraged us and been helpful during the revision of our manuscript. Thank you so much.

Appendix B

Dear Dr. Abe,

as the associate editor handling your submission, I have now received comments from one of your previous reviewers. Based on these comments and my own reading of the manuscript, I will suggest to my editor-in-chief accept your paper for publication pending minor revisions. Your reviewer has made some very helpful and concrete comments on how to further improve the quality of your contribution. I agree with the reviewer that the minor changes will help to more closely guide the reader and highlight the specific novelty of your work.

We are looking forward to receiving a revised version at your earliest convenience.

With kind regards,

Oliver Schülke

Dear Dr. Oliver Schülke (Associate Editor),

We would like to resubmit the revised version of our manuscript entitled “Bipartite network analysis of ant-task associations reveals task groups and absence of colonial daily activity”. We deeply thank the reviewer for thoroughly reviewing our manuscript and making many helpful comments.

Our point-by-point responses to the reviewer’s comments are detailed below. The modified parts of the manuscript are marked in blue. Also, we found a minor mistake in Figure S3 and improved it. It does not affect our conclusions.

Sincerely yours,

Masato S. Abe

RIKEN Center for Advanced Intelligence Projects,
Nihonbashi 1-chome Mitsui Building, 15th floor,
1-4-1 Nihonbashi, Chuo-ku, Tokyo 103-0027, Japan
Tel: +81-048-467-3627
masato.abe@riken.jp

Response to Reviewer 2

Comments to the Author(s)

Overall I think this paper is much improved. It is clear that the authors are building on Pasquaretta and Jeanson by first examining nestedness in the bipartite individual-task network and then expanding the bipartite method by (a) considering task-task module networks and including it in measurement of individual-level specialization, and (b) considering individual differences (in age, body size, etc.) within this bipartite network.

Some of the findings in this paper are very interesting. The ability of your metric to detect possible signs of system robustness (absence/presence of nestedness) and to detect functional specialization (lines 385-386) are very nice from a complex systems perspective.

Response: We strongly appreciate for your careful reading and many helpful comments on our manuscript. We have modified my paper appropriately to address your concerns. Below, we give a point-by-point reply to each criticism. Please note that the improved parts in the manuscript are indicated in blue.

Comment 1:

Thus, I would suggest engaging a bit more with complex systems literature (either the portion focused on social insects or the broader literature), since you clearly touch on these themes but mostly in passing. Your metrics seems to offer interesting insights into these aspects of colony structure, and you could well lean into it a bit more. You may also wish to engage with (or at least acknowledge) some of the older social insect literature that looked at task networks but constructed them largely manually (e.g., Deborah Gordon's earlier work on task flow networks in harvester ants, or the figure in Fewell 2003 in Science). I see some similarities in your network module approach, although yours is obviously far more modern.

Unfortunately, there are still some clarity issues with the writing. As you'll see below there are still portions of the paper where it is hard for the reader to clearly follow the logic and message of the paper. My stylistic suggestions do not need to be followed but instead are intended to point out where I found it hard to follow and provide a possible way to improve (not because I believe my way is best but simply to try to be helpful). More time dedicated to truly polishing the writing will bring this paper into final form!

Response 1: Thank you for the suggestion. We have cited some papers and revised the introduction (lines 46-49, 53).

We agree to that they laid the groundwork of the task network. We have added three papers (Gordon 1989; 1996; Fewell 2003, lines 93-95).

Comment 2:

INTRODUCTION

The introduction now makes it clear that this paper will make methodological contributions to the field. That said, the new sections that were added were still a bit hard to parse. For example, line 74-79 has an abrupt transition from discussing the gap in the Pasquaretta and Jeanson method that still needs to be filled to "Also, we focused on a network structure and incorporating other biological features (e.g., diurnal activity, age)." The use of "also" here is confusing relative to the previous sentence; you haven't discussed what you have done up until this point so "also we focused" is confusing. The end of this paragraph is key for the reader to understand what the gap is and how you intend to fill it before discussing what exactly you did in the following paragraphs. Please revise this carefully to help the reader along.

The following four new paragraphs where you describe specifically the four core things you are building upon the bipartite method can be a bit hard to follow. As of now it felt a bit like you are listing a bunch of things you will do/discuss in this paper but it's not clear why that's the case. Maybe if you could provide a roadmap (e.g., a numbered list of the four main things you're doing) to the reader before these paragraphs where you succinctly describe the four ways in which you will be building upon the Pasquaretta and Jeanson method, it will be much easier to follow. It doesn't have to be a roadmap, but consider now that unless the reader hops from topic sentence to topic sentence in this section, one can get a bit lost in the discussion and lose sight of the exact novelty you are bringing to the table.

Response 2: Thank you for pointing this out. Yes. It was not good writing. We have revised the points you suggested. We also made a numbered list of the four issues we are solving (lines 77-81).

Comment 3:

METHODS

Lines 229-232: I think you could explain more about why exactly you want to compare your c-score metric with d'. You mention that you are going to compare your metric to d', but why? As of now it just seems an arbitrary comparison, so please make clear to the reader why you wish to compare it so that the later results showing the difference between the two are much clearer. Are you comparing it because you believe the two metrics will capture different aspects of division of labor/bipartite

network structure? Are you comparing it because your work is expanding on the study that introduced d' so you want to use it as a baseline?

Response 3: Thank you for your helpful comments. We compared two metrics to investigate whether the two metrics capture different aspects of the bipartite network structure. We have added the explanation (lines 235-239).

Comment 4:

DISCUSSION

Lines 374-378: This paragraph seems to be a result and should probably be entirely/mostly moved to the results section. It mostly seems to be describing an empirical finding and not actually adding new discussion/insight to the finding.

Response 4: Yes. We have moved this paragraph to the result section (lines 293-297).

Comment 5:

Lines 385-386: "The fact that some complete specialists with c -score = 0 have d' indiv < 1 (figure S2) reflects that the c -score can identify specialists from a functional point of view." In my opinion this is a key finding of your method and should be highlighted more. You are showing how your method and metrics gain a new insight into division of labor that is missed by other measurements.

Response 5: Thank you for your comment. We have added explanation about what c -score reflect in task allocation network (lines 396-400).

Comment 6:

As with the above lines, in general in your discussion, I think you could lead more with a key finding or point and discuss that instead of summarizing what you did in your study (which you have already done in the methods and results section!). You have some interesting findings that deserve highlighting and discussion but some of them are buried within your discussion paragraphs instead of leading them! For example, it seems the key topic in paragraph 2 of the discussion is your findings on nestedness, group size, and its implications for robustness, but it is sort of buried later in the paragraph. You could advocate more clearly in your paragraph that your approach gives insights into the organization of colonies that may be missed without your approach. I haven't heard many

others discuss how the nestedness of task structure may change with group size, adding to your understanding of scaling effects in colonies. Very interesting!

Response 6: Thank you for your comment. We have revised this paragraph (lines 351-364). We moved a sentence to the result section (lines 262-263). We discussed our idea of the relationship between nestedness and colony size (lines 357-364).